# Tanning Bed Legislation for Minors: A Comprehensive International Comparison

**DOI:** 10.3390/children9060768

**Published:** 2022-05-24

**Authors:** Katharina Diehl, Karla S. Lindwedel, Sonja Mathes, Tatiana Görig, Olaf Gefeller

**Affiliations:** 1Department of Medical Informatics, Biometry and Epidemiology, Friedrich-Alexander-Universität Erlangen-Nürnberg, 91054 Erlangen, Germany; katharina.diehl@fau.de (K.D.); karla.lindwedel@fau.de (K.S.L.); tatiana.goerig@fau.de (T.G.); 2Department of Dermatology and Allergy, Technische Universität München, 80802 München, Germany; sonja.mathes@mri.tum.de

**Keywords:** tanning bed, sunbed, legislation, ban, adolescent, minors, ultraviolet radiation

## Abstract

Tanning beds have been classified as carcinogenic to humans. As a result, many countries have enacted laws regulating the use of commercial tanning beds, including bans for minors. However, there is no international overview of the current legal status of access restrictions for minors that provides details on their specific design regarding age limits and possible exceptions to the statutory regulation. Therefore, we performed a comprehensive web search of current tanning bed legislation for minors on the three continents North America, Australia, and Europe. Our findings regarding the existence and concrete design of access restrictions are presented graphically, using maps. We found a wide variety of different legislations. In Australia, a total ban on tanning beds exists, while in New Zealand, tanning bed use is banned for minors. In Europe, about half of the countries have implemented a strict ban for minors. In North America, we found differences in the age limit for access restrictions between the states, provinces, and territories for those regions that implemented a ban for minors. In the United States, some states have rather “soft bans” that allow use by minors with different types of parental consent. The patchwork in legislation calls for harmonization. Therefore, our comparison is an important starting point for institutions such as the World Health Organization or the European Commission to advance their goals toward a harmonization of tanning bed legislation in general and for minors in particular.

## 1. Introduction

Nowadays, tanned skin is not dependent on the weather, the season, or the latitude where a person lives. People who aim for a tanned appearance can fulfill their desire any-time by using tanning devices emitting artificial ultraviolet (UV) radiation. Outside the equatorial regions, the tanning industry operates a wide range of indoor tanning salons, offering different facilities such as tanning beds, tanning lamps, or tanning booths. The precursors of these artificial tanning devices date back to the end of the 19th century. In 1895, the Danish physician Niels Ryberg Finsen developed the “Finsen light”, a special form of a carbon arc lamp [1]. UV radiation from a carbon arc was condensed through four water-cooled tubes fitted with quartz lenses. The Finsen light was successfully applied to treat lupus vulgaris. Its inventor received the Nobel Prize in Medicine and Physiology for his work in 1903 and is nowadays viewed as the father of modern phototherapy [2]. Only some years after Finsen, the German engineer Richard Küch and his colleague Tscheslaw Stefan Retschinsky, working for the German company Heraeus, developed a method to produce high-purity quartz glass from molten mountain crystal and used it to build a high-pressure quartz lamp. Heraeus manufactured these sunlamps on an industrial scale, and they were used for decades in the phototherapy of acne and other skin diseases [3]. The modern era of artificial tanning devices began when the German engineer Friedrich Wolff developed the first tubular low-pressure UV lamp in the 1970s. Wolf’s original intention was to treat seasonal affective disorders by using artificial UV radiation resembling the sun. However, tanned skin as a side effect of the treatment quickly became the main reason for demand for such treatment, showing the market potential of artificial tanning devices outside of medical applications [4].

Indoor tanning in tanning salons grew in popularity in the 1980s and 1990s [5,6,7,8]. The tanning industry’s efforts to convince the public that indoor tanning increases attractiveness and health were successful [9], despite growing scientific concerns about skin cancer risks associated with such behavior [10]. Early epidemiologic studies addressing the melanoma risk of indoor tanning [11,12,13,14,15,16], as well as laboratory data and animal experiments focusing on the role of ultraviolet A radiation in skin cancerogenesis [17,18,19], provided accumulating evidence of the harmful health effects of indoor tanning.

Around the turn of the millennium, the attitude of the health authorities began to change, and the debate about more regulation for tanning devices gained momentum. In 2003, the World Health Organization (WHO) recommended that governments should introduce comprehensive legislation that regulates the operation of tanning beds in a legally binding manner and allows for on-site enforcement [20]. During the following years, several countries tightened their regulations up to the point of a total or partial ban of tanning salons. Minors, in particular, were affected by legal bans or access restrictions to tanning beds. However, international regulatory measures introduced in the last decades varied considerably, with the result that regulations in different countries are now very heterogeneous. 

The aim of this paper is to present the current state of international regulations on access of minors to tanning beds. For this purpose, we carried out a comprehensive investigation of the official legislation in all countries with populations predominantly consisting of fair-skinned people. In this paper, we report results from all 47 European countries; the two North American countries, the United States (US) and Canada (including local regulations in all 50 US states and all 13 Canadian provinces and territories); and Australia and New Zealand.

## 2. Materials and Methods

We conducted a comprehensive web search regarding legislation on tanning bed use on the three continents North America, Australia, and Europe. The focus of this manuscript is on whether there exists legislation regarding tanning bed use in minors. The main sources of information were legislative texts of the specific countries.

For the search, we used the Google search engine in the Chrome browser. The browser allowed for a direct translation into different languages. We used the respective national languages for the search based on the terms “tanning bed”, “legislation”, and the country’s name. If there were no government web pages among the first hits, the search was extended by the term “site:.gov.*country code*”. To identify the exact name of the legislation, online newspaper articles were also used for a deeper search.

The identified legislation texts were translated into German language by using Google translate to be better understandable and comparable. The information regarding existing legislation was entered into an Excel spreadsheet. When information was missing, a more detailed search using the Google search engine was performed. Search activities took place between 28 February 2021 and 16 January 2022. Maps were created with Microsoft Powerpoint (version 2019) based on maps supported by Bing.

### 2.1. Exceptions to the General Search Procedure

Exceptions to the general procedure regarding the web search were the US and Canada. For the US, we used the website of the AIM at Melanoma Foundation (https://www.aimatmelanoma.org/legislation-policy-advocacy/indoor-tanning/, accessed on 16 January 2022) as the starting point for further search. For Canada, the recent article by Gosselin and McWhirter [21] served as our starting point, as it provided a good overview on the specific names of the corresponding tanning bed legislations.

For those countries where we were unable to locate a source of official regulatory information during our web search, we established, whenever possible, personal contact with government agencies, national health institutes, and embassies of these countries in Germany to determine whether any restrictions existed for minors.

### 2.2. Presentation of Findings

Details regarding the specific legislation—including the type of access restrictions and the link to the online information on legislation—in all countries can be found in the Appendix A (Table A1). The created maps give a graphical overview about the three included continents. To visualize the different types of tanning bed access restrictions, we used an identical color code in all figures, allowing us to distinguish between (1) a total ban for all ages, (2) a strict ban for minors under the age of 18, (3) a strict ban for minors with a different age limit, (4) a soft ban for minors implementing different barriers to tanning bed access, and (5) no restrictions for minors.

## 3. Results

In Australia, a total ban on commercial tanning beds was introduced in all Australian states and territories, except the Northern Territory, where there are no commercial tanning salons (Figure 1). For New Zealand, we identified a strict ban for minors under the age of 18.

In Europe, no country has implemented a total ban on tanning beds comparable to Australia. However, 25 out of 47 countries have implemented strict access restrictions for minors. For the United Kingdom, we found a strict ban for minors in all four countries: England, Northern Ireland, Scotland, and Wales. 

As shown in Figure 2, the countries with restrictions for minors introduced a strict ban for using tanning beds for individuals under the age of 18. The sole exception is Latvia, where the age limit is not explicitly defined. For Bosnia Herzegovina, we used data for the Republic of Srpska representative for the whole country.

In eight of the 22 countries without access restrictions for minors, legislation regarding tanning beds exists but does not include any restrictions for minors. In six countries, there are superordinate regulations concerning tanning beds—for instance, regarding irradiation intensity and wavelengths via the European Committee for Electrotechnical Standardization (CENELEC)—but no further country-specific legislation. Eight countries do not have any legislation on tanning bed use, including, amongst others, Russia, where a ban for minors was part of a comprehensive legislation on tanning bed use that was in law from 2010 but was annulled in 2021.

In the US, we found more variation in legislation for tanning bed use by minors (Figure 3): while five states have no restrictions, 45 have introduced restrictions. In those states without restrictions for minors, the guidelines of the Food and Drug Administration (FDA) are valid regarding, for instance, warning signs. Two of the states do have additional legislation on tanning bed use; they, do not include any restriction for minors.

Besides a strict ban for those under the age of 18 (*n* = 23), some states have chosen different age limits for their strict ban (*n* = 9). New York is currently debating a senate bill that contains a ban for under-21-year-olds, which would mean an increase in the current age limit of 18 years.

Twenty states have introduced a rather “soft ban” or a combination of strict and soft legislation depending on age: among these 20 states, seven states have a strict ban for younger adolescents and a soft ban comprising some form of access restrictions for older adolescents. Access restrictions mean that minors are only allowed to use tanning facilities when parents or legal guardians accompany the minor (*n* = 1), when parents or a legal guardian show up at the tanning salon to sign a consent form (*n* = 2), when minors present written parental consent (*n* = 3), or a combination of these three barriers, depending on the minor’s age (*n* = 1). 

Eight states have introduced a soft ban with parents’ or legal guardians’ presence at the tanning salon to sign a consent form (*n* = 7) or on the condition that minors present written parental consent (*n* = 1). 

The remaining five states have combined different components of soft-ban barriers depending on the minor’s age: While two states require parental accompaniment for younger adolescents, written parental consent is sufficient for older adolescents. Two further states require parental accompaniment for younger adolescents and the presence of parents to sign consent for the older ones. Finally, one state requires the presence of a parent to sign consent in younger adolescents, while it is sufficient for older adolescents to present written parental consent.

In Canada, two territories do not have any restrictions for minors, although they have additional legislation on tanning bed use that applies to all of Canada. All other provinces and territories (*n* = 10) have introduced a strict ban for minors (Figure 4). The age limit for the ban varied between 18 years of age (*n* = 6) and 19 years of age (*n* = 4).

## 4. Discussion

In 2009, the International Agency for Research on Cancer (IARC), an intergovernmental agency that is part of the WHO, classified artificial UV radiation as carcinogenic to humans and placed it in the group I of those carcinogens with the strongest evidence for carcinogenicity [22], following the recommendation of the IARC Working Group on Artificial UV Light and Skin Cancer [23]. Several later meta-analyses [24,25,26] corroborated that tanning bed use is associated with a significant increase in the risk of developing cutaneous melanoma and keratinocyte carcinoma. To prevent individuals from potential skin cancer risks from using tanning beds, the WHO suggested two options: either to ban tanning beds or to restrict and manage the use of tanning beds, accompanied with education and information on the subject [27]. 

Our comprehensive search for international legislation regarding tanning bed use among minors revealed a high variation in the drafting of legislation on the three continents. Existing legislation ranged from a total ban in Australia to strict bans on minors: for instance, while there are different age limits and more softened bans that implement access barriers but allow minors to use tanning beds if their guardians provide a signed consent form or accompany the minor, there are also countries where no regulations on the use of tanning beds among minors exist. However, it should be noted that there may be countries or territories where there are no commercial tanning beds at all and, therefore, no legal regulation is necessary. While not all countries may officially state that this is the case, for the Northern Territory of Australia, this information was provided so that this area was marked as if there is a total ban, because, by law, no commercial tanning beds may be operated in the future. When this was not clearly stated for a certain area or country and we could not identify any information on restrictions regarding minors, these countries were classified as “no existing legislation for minors”.

It should be emphasized that children and adolescents are an especially vulnerable group that needs tailored concepts for prevention and protection from UV radiation [28,29,30]. Although the harmful effects of UV radiation on human skin are not limited to children and adolescents [31], in this age group, the time spent outdoors is, on average, longer than for adults [32]. This results in a higher cumulative dose of UV radiation per year of life in this age range than in later years of life [32]. In addition, children and adolescents have, on average, a much longer remaining lifespan than adults, so that UV-radiation-triggered processes of skin carcinogenesis, which has a long latency period, are more likely to manifest in them during later life than in those in whom these processes are initiated at an older age [30]. Overexposure of UV radiation during childhood has been found consistently to be associated with higher nevus density [33], which, in turn, is strongly related to the risk of developing melanoma later in live [32,34]. From a prevention perspective, it therefore seems reasonable to restrict the use of tanning beds for minors so that they are not exposed to artificial UV radiation in addition to UV radiation from natural sunlight.

Previous research has shown, however, that minors find loopholes to use tanning beds, despite a legal ban [35,36,37,38,39]. However, data from some countries also revealed that tanning bed use in minors has decreased over time since the implementation of the ban [35,40,41]. As we have seen in the US, several states allow commercial tanning bed use for minors when they have parental permission (consent or accompaniment). A study by Guy et al. [42] provided evidence that laws including strict age restrictions for tanning bed use may be more effective in reducing their use by minors than parental-permission laws without an age restriction, because the latter ones were not associated with a lower prevalence of indoor tanning. 

An alternative concept to bans is the idea of so-called “buy-back” schemes that encourage the removal of tanning beds, while tanning bed operators are financially compensated for the enforced changes to their business [35]. An Australian study, for instance, showed that, after having banned commercial tanning beds, the interest in indoor tanning among the population has fallen, while the interest in spray tanning has remained high [43]. Spray tanning, also known as sunless tanning, is often seen as “healthy alternative” to indoor tanning especially for those with a pale skin phenotype [44]. Thus, promoting spray tanning could be a strategy to reduce the demand for tanning beds.

Our investigation of international legislation on tanning beds with respect to access restrictions for minors is not the first of its kind. In 2003, Delavalle et al. [45] reviewed access restrictions for minors in English-speaking countries (US, Canada, Australia, New Zealand, and the UK), as well as in France, and compared it with tobacco restrictions for minors in the same countries. In 2005 and 2007, the same group published short updates on changes in tanning bed legislation for minors in the US [46,47]. A comprehensive review of international legislation restricting access to indoor tanning was performed by Pawlak et al. [48]. Their paper from 2012 represented the legal status quo of 2011 and is already outdated today, as many countries have changed their relevant legislation over the past decade. In 2019, Longo et al. [49] described the legislative situation for tanning bed use in Europe. Their approach to assess the relevant data was, however, completely different from ours. They used a questionnaire specifically designed to capture details of the regulatory framework for tanning bed use and sent the questionnaire to all country coordinators of Euromelanoma, a network of scientists in Europe with a common interest in melanoma that was founded in 1999 and is now active in 33 countries. The assessment via a questionnaire was performed between April and December 2012, and an update—using a methodology not described in the publication—took place in 2017. As the response rate was only 64% and not all European countries were represented in the Euromelanoma network, the survey had some gaps, and the report could only describe the situation in 23 countries. Also focusing on Europe, del Colombo and Vasile [50] recently reviewed the actions taken by the European Commission to improve the safety of tanning bed use. They described the process of deriving standardization in the CENELEC context and political measures supporting the development. Rodriguez-Acevedo et al. [51] incorporated a world map of indoor-tanning restrictions in a recent systematic review and meta-analysis aimed at analyzing changes in region-specific indoor tanning prevalence after indoor tanning devices were classified as carcinogenic by the IARC in 2009. The authors did not provide a detailed explanation describing how the information leading to the map was compiled. They only gave a reference to a WHO document from 2017 on public health management of artificial tanning devices [27] that lacks, however, detailed information on country-specific access restrictions. Overall, the information provided by Rodriguez-Acevedo et al. [51] corresponds quite well with findings from our investigation that can be viewed as an update utilizing a transparent methodology.

Although we performed an extensive search for international legislation in all countries of the study region, we cannot guarantee that there is no further information on legislation that we have not been able to include in this comparison. It is conceivable that not all countries make their legal regulations available on the internet or that we were unable to locate the corresponding website, e.g., due to language barriers. In eight European countries, our search did not reveal any current legislation on tanning beds. In these cases, we tried to confirm the non-existence of tanning bed legislation by contacting the health authorities and embassies of these countries or by searching for additional media articles on the tanning bed industry in these countries. The success of these activities was limited. Only for three of the eight countries were we able to obtain independent verification of the non-existence of tanning bed regulation. For another six European countries, our search was unable to locate country-specific tanning bed legislations, but these countries have at least adopted the technical safety specifications for tanning devices operated in CENELEC member states, as well as the European Committee for Standardization’s (CEN) regulations for the training of personnel working in tanning salons. For four of these six countries, we obtained confirmation that no specific tanning bed legislation is in place. Although we incorporated countries from three continents in our investigation, our comparison does not cover the entire world. We left out Asia and Africa completely as we expected to find country-specific legislation on tanning bed use only in very few countries, such as, for example, the Republic of South Africa, where a liberal regulation imposing no access restrictions for minors exists [52]. The high expense of workload to verify the non-existence of legislation in these regions did not stand in adequate relation to the limited yield of information. Due to language barriers, we also did not search systematically for legislation in South America, acknowledging that Brazil became the first country in the world to ban the trade and use of tanning beds in 2009 [53,54]. It is also worth noting that we have focused only on access restrictions for minors in this manuscript. We did not look in detail at the terms of access (including, for instance, mandatory protective eyewear), warning signs, restricted advertising, and educational requirements. In addition, technical regulations for tanning devices differ between countries [55], and we also have not discussed this in our manuscript.

## 5. Conclusions

Our international comparison regarding regulations on the use of commercial tanning beds for minors showed a large variety in legislation. There is no harmonization between countries, for instance, in the European Union, or even within a country, for example, in the US. However, a harmonization of legislation on tanning bed use by minors, as well as tanning bed use in general, may strengthen the efforts to reduce skin cancer incidence in the long run. Our findings are an important basis for further work of the WHO, but also of the EU Commission. 

The WHO’s aim is to either ban tanning beds or to combine restrictions with managing tanning beds and informing users [27]. The maps provided in this paper will foster transparency about the status quo of legislation in the different countries and regions to identify progress in legislation and to derive needs for further action. In addition, our findings fit well into the schedule of the European Commission’s “Europe’s Beating Cancer Plan”. For 2023, the European Commission is scheduled to explore measures to prevent exposure from tanning beds [56]. Our overview will support their work, which should aim to ensure that member states cooperate on this topic and implement common legislation on commercial tanning bed use by minors.

## Figures and Tables

**Figure 1 children-09-00768-f001:**
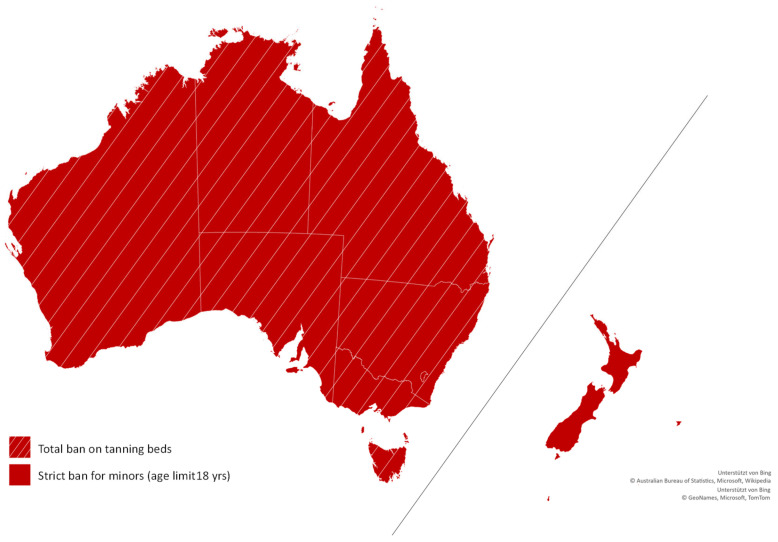
Legislation on tanning bed use for minors in Australia and New Zealand. Map supported by Bing. Copyright by GeoNames, Microsoft, TomTom, Australian Bureau of Statistics, Microsoft, and Wikipedia.

**Figure 2 children-09-00768-f002:**
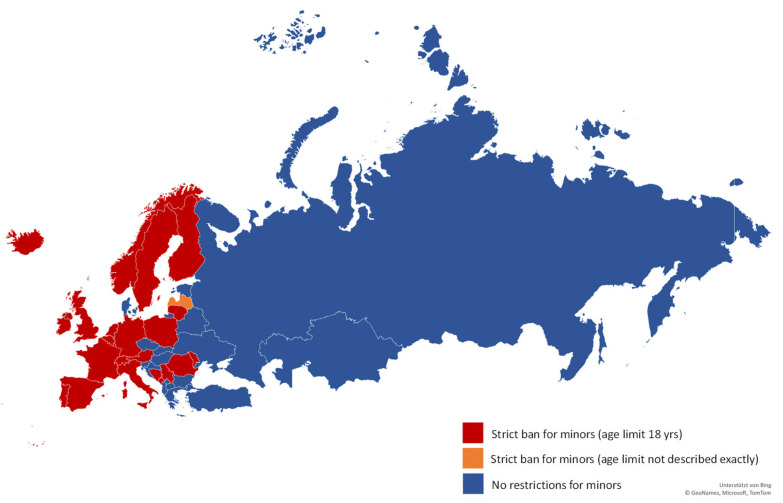
Legislation on tanning bed use for minors in Europe. Map supported by Bing, Copyright by GeoNames, Microsoft, and TomTom.

**Figure 3 children-09-00768-f003:**
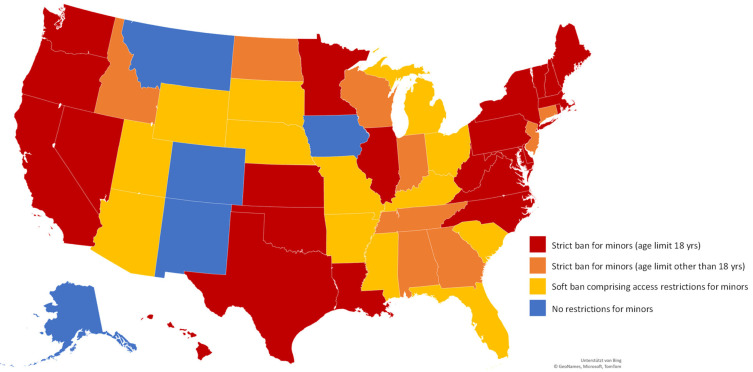
Legislation on tanning bed use for minors in the United States. Map supported by Bing, Copyright by GeoNames, Microsoft, TomTom.

**Figure 4 children-09-00768-f004:**
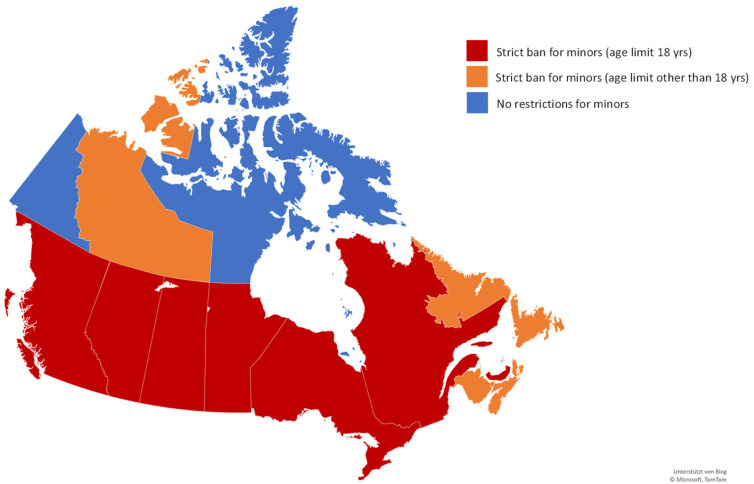
Legislation on tanning bed use for minors in Canada. Map supported by Bing, Copyright by Microsoft, and TomTom.

## Data Availability

Not applicable.

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
