# Peer review of "Tanning Bed Legislation for Minors: A Comprehensive International Comparison"

_children, 2022, doi:10.3390/children9060768_

Round 1

Reviewer 1 Report

Children - 1726221 Tanning Bed Legislation for Minors: A Comprehensive International Comparison

This submission is a  descriptive comparison of tanning bed legislation across and within countries with large fair-skinned populations:  Australia, New Zealand, Europe and North America.

This submission is a well-documented, written and offers important insight to the current status of tanning bed legislation aimed at minors in the countries research. The authors provide color-coded maps representative of the levels – if any- of legislation for tanning facility use among minors. While the color coding is written in each figure descriptor, an actual legend of the color coding would ensure clarity, even if reprints are available only in monochromic settings.

The only additional suggestion may be the inclusion of a single table that lists each country individually and summarizes all the information provided in the map. This table would allow clear notation of the countries (six European) for which no specific legislation could be found.  Although I believe this table would be helpful to others, it is merely a suggestion.

This manuscript as written presents the methodology and outcome of the investigators collection and comparison of tanning bed legislation aimed at minors clearly. This submission will be a valuable contribution to the current literature.

Author Response

We thank the reviewer for the positive evaluation of our manuscript. We appreciate the remarks to enhance the presentation of findings in the four figures and the accompanying table. Reacting to the suggestions we have

(i) incorporated legends explaining the color code in all four figures and eliminated the description of the color code in the text to ensure better readability of the figures

(ii) added the information that no legislation is in force for the corresponding countries to Table A1. In the revised form the Table A1 gives all country-specific details on tanning bed legislation and supplements the figures that cannot capture all details on the heterogenous access restrictions.

Once again, we thank the reviewer for his thoughtful and constructive criticism that helped to improve the presentation of our results.

Reviewer 2 Report

The study:”Tanning Bed Legislation for Minors: A Comprehensive International Comparison” ” is very interesting work. Authors found very useful topic, especially that tanning is very popular among young and adults, and becoming more and more popular among children. There is an ample of evidence on different tanning devices and possible problems that stem from their use, especially harmful effect on skin (an example is 2016 BJD article DOI 10.1111/bjd.14388). There is insufficient data depicting legislative side for the use of tanning beds.  It is very important to have data, such as presented here, to make people aware of the possible problems that tanning devices might cause as a growing public health problem worldwide.

Author Response

We thank the reviewer for the positive evaluation of our manuscript. We appreciated the comments on the importance of our work. Since the reviewer did not suggest any changes, we have no specific explanation on what has changed in the revised version in response to the comments to share.